



# Attributing correlation skill of dynamical global precipitation forecasts to statistical ENSO teleconnection using a set theory based approach

Tongtiegang Zhao[1], Haoling Chen[1] and Quanxi Shao[2]

[1] Center of Water Resources and Environment, Southern Marine Science and Engineering Guangdong Laboratory (Zhuhai), School of Civil Engineering, Sun Yat-Sen University, Guangzhou, China
[2] CSIRO Data61, Australian Resources Research Centre, Bentley, WA, Australia

*Correspondence to*: Tongtiegang Zhao (zhaottg@mail.sysu.edu.cn)

**Abstract.** Climate teleconnections are essential for the verification of valuable precipitation forecasts generated by global climate models (GCMs). This paper develops a novel approach to attributing correlation skill of dynamical GCM forecasts to statistical El Niño-Southern Oscillation (ENSO) teleconnection by using the coefficient of determination ($R^2$). Specifically, observed precipitation is respectively regressed against GCM forecasts, Niño3.4 and both of them and then the intersection operation is implemented to quantify the overlapping $R^2$ for GCM forecasts and Niño3.4. The significance of overlapping $R^2$ and the sign of ENSO teleconnection facilitate three cases of attribution, i.e., significantly positive anomaly correlation attributable to positive ENSO teleconnection, attributable to negative ENSO teleconnection and not attributable to ENSO teleconnection. A case study is devised for the Climate Forecast System version 2 (CFSv2) seasonal forecasts of global precipitation. For grid cells around the world, the ratio of significantly positive anomaly correlation attributable to positive (negative) ENSO teleconnection is respectively 10.8% (11.7%) in December-January-February (DJF), 7.1% (7.3%) in March-April-May (MAM), 6.3% (7.4%) in June-July-August (JJA) and 7.0% (14.3%) in September-October-November (SON). The results not only confirm the prominent contributions of ENSO teleconnection to GCM forecasts, but also present spatial plots of regions where significantly positive anomaly correlation is subject to positive ENSO teleconnection, negative ENSO teleconnection and teleconnections other than ENSO. Overall, the proposed attribution approach can serve as an effective tool to investigate the source of predictability for GCM seasonal forecasts of global precipitation.

## 1 Introduction

Precipitation is one of the most important hydrological variables and is an integral part of the global water cycle (Scofield & Kuligowski, 2003; Huffman et al., 2007; Ushio et al., 2009; Schneider et al., 2016; Beck et al., 2019). It plays a key role in driving hydrological processes at catchment, regional and continental scales (e.g., Robertson et al., 2013; Wu et al., 2014; Yuan et al., 2014; Greuell et al., 2018; Lakshmi & Satyanarayana, 2019). Despite the importance, the forecasting of precipitation remains a formidable task due to complex interactions of ocean, atmosphere and land surface processes





(Doblas-Reyes et al., 2013; Vano et al., 2014; Johnson et al., 2019; Zhao et al., 2019; Tesfa et al., 2020). Comparing multiple sets of global temperature and precipitation forecasts from the North American Multi-Model Ensemble (NMME) experiment, Becker et al. (2020) highlighted that there are substantial improvements in temperature forecasts over both land and ocean during the past decades and that there is still plenty of room for improvement of global precipitation forecasts.

  Global climate models (GCMs) generate valuable forecasts of worldwide precipitation for hydrological modelling and water
management (Doblas-Reyes et al., 2013; Kirtman et al., 2014; Schepen et al., 2020). Nowadays, GCMs have been employed by major climate centers around the world to produce operational climate outlooks (Demargne et al., 2014; Delworth et al., 2020). For example, the Climate Forecast System version 2 (CFSv2) of the U.S. National Centers for Environmental Prediction (NCEP) has been implemented for coupled ocean-atmosphere forecasting since 2011 (Saha et al., 2010); the European Centre for Medium-Range Weather Forecasts (ECMWF) System 5 model became operational in 2017 (Johnson et
al., 2019); and the Seamless System for Prediction and Earth System Research (SPEAR) became the next generation modelling system at the Geophysical Fluid Dynamics Laboratory (GFDL) in 2020 (Delworth et al., 2020). In the meantime, GCM forecasts have been increasingly incorporated into forecasting systems of streamflow, crop yield and soil water and they are shown to create enormous socioeconomic benefits (e.g., Vano et al., 2014; Peng et al., 2018; Wang et al., 2019).

  Climate teleconnections, which are widely used in conventional statistical hydrological forecasting (Lima & Lall, 2010;
Steinschneider & Lall, 2016; Mendoza et al., 2017; Mortensen et al., 2018; Wang et al., 2020), are an essential part in assessing the skill of GCM forecasts (Neelin & Langenbrunner, 2013). That is, a number of teleconnection patterns are usually investigated upon the issuance of a new set of GCM forecasts (Kim et al., 2012; Jia et al., 2015; Delworth et al., 2020). For example, El Niño-Southern Oscillation (ENSO) and Madden-Julian oscillation (MJO) have been investigated for CFSv2 forecasts (Saha et al., 2014). North Atlantic Oscillation (NAO), Arctic Oscillation (AO) and Pacific North American
(PNA) teleconnections have been assessed for Global Earth Observing System (GEOS) modeling and data assimilation system (Molod et al., 2020). ENSO and Pacific decadal oscillation (PDO) have been examined for GFDL-SPEAR forecasts (Delworth et al., 2020). It is generally found that skillful GCM forecasts are owing to effective formulations of teleconnection patterns (e.g., Molteni et al., 2011; Merryfield et al., 2013; Saha et al., 2014; Jia et al., 2015; Delworth et al., 2020).

There have been in-depth investigations of ENSO for GCM forecasts as it is one of the most prominent modes of climate variability (Fu et al., 1997; Wang et al., 2003; Feng & Hao, 2021). Attention is usually paid to regions subject to prominent ENSO influences (Kim et al., 2016; Rivera & Arnould, 2020; Vashisht et al., 2021). For instance, possible contributions of ENSO to precipitation forecast skill were investigated for the west coast of North America (Pegion & Kumar, 2013; Chen & Kumar, 2020). The influence of ENSO on the East Asian-western Pacific climate was studied for CFSv2 forecasts (Yang &
Jiang, 2014) and also for climate projections in the Coupled Model Inter-comparison Project Phase 5 (Gong et al., 2015; Kim et al., 2016; Kim & Kug, 2018). Process-based evaluation linking ENSO to summer precipitation was performed over eastern Africa (Vashisht et al., 2021). Nevertheless, at the global scale, the relationship between ENSO teleconnection and GCM precipitation forecasts is yet to be illustrated.





This paper is devoted to attributing correlation skill of dynamical CFSv2 forecasts (e.g., Saha et al., 2010; Yuan et al., 2011; Jia et al., 2015; Becker et al., 2020; Zhao et al., 2020a) to statistical ENSO teleconnection at the global scale. A novel approach based on the coefficient of determination ($R^2$), which measures the ratio of the explained variance to the total variance, is devised to facilitate the attribution. The significance of overlapping $R^2$ is tested by bootstrapping in order to identify where correlation skill is attributable to ENSO teleconnection. The novelty of the approach is the implementation of classical set operations through simple linear regression. As will be demonstrated through the case study of CFSv2 forecasts, three cases are effectively revealed: (1) significantly positive anomaly correlation attributable to positive ENSO teleconnection, (2) significantly positive anomaly correlation attributable to negative ENSO teleconnection and (3) significantly positive anomaly correlation not attributable to ENSO teleconnection.

## 2 Data description

GCM forecasts comprise a typical high-dimensional dataset (Kirtman et al., 2014; Saha et al., 2014; Chen & Kumar, 2016; Becker et al., 2020; Zhao et al., 2020b). For the CFSv2 forecasts investigated in this paper, there are five dimensions: (1) forecast start time $s$; (2) lead time $l$; (3) ensemble size $n$; (4) latitude $y$; and (5) longitude $x$. $s$ represents the number of months since the benchmark time that is January 1982; $l$ is the number of months ahead, which ranges from 0 to 9 month for the CFSv2 forecasts; $n$ =1, …, 24, i.e., the total number of ensemble members is 24; $y$ ranges from −90 to 90 while $x$ is from 0 to 359, with a horizontal resolution of 1.0 °latitude by 1.0 °longitude. The set of forecasts is denoted by

$$F = \left[ f_{s,l,n,y,x} \right], \tag{1}$$

in which $f$ represents forecast values specified by the five dimensions and $F$ is the dataset of forecasts.

There are three dimensions for the dataset of observed precipitation corresponding to forecasts (Xie et al., 2007; Infanti & Kirtman, 2015; Schneider et al., 2016). They are target time $t$, which is equal to the sum of start time $s$ and lead time $l$ to align observations with forecasts; latitude $y$; and longitude $x$. The set of observed precipitation corresponding to the forecasts is denoted as

$$O = \left[ o_{t,y,x} \right] \quad (t = s+l). \tag{2}$$

The CPC global daily Unified Rain-gauge Database (CPC-URD), which has been widely used in the analysis of regional and global precipitation (Xie et al., 2010), is used as the referenced observed precipitation.

The correlation skill, which is in the form of the Pearson's correlation coefficient, is calculated so as to relate CFSv2 precipitation forecasts to CPC-URD observations:





$$r(o, f) = \frac{\sum_k (o_k - \overline{o})(f_k - \overline{f})}{\sqrt{\sum_k (o_k - \overline{o})^2} \sqrt{\sum_k (f_k - \overline{f})^2}}.$$

(3)

In Eq. (3), $k$ represents the target year ($k=1982, 1983, \ldots, 2010$), where other dimensions of $o$ and $f$ are omitted for the sake of simplicity. The value of $r(o, f)$ measures how well CFSv2 forecasts correspond to observed precipitation. A significantly positive $r(o, f)$ implies that large (small) forecasts are indicative of large (small) observations, whereas a neutral or negative $r(o, f)$ indicates non-skillful forecasts (Yuan et al., 2011; Zhao et al., 2020a; Zhao et al., 2020b).

It is noted that forecasts/observations, which are monthly, are aggregated into seasonal. The aggregation is meant to facilitate the analysis by season. The attention is paid to the latest forecasts. That is, seasonal precipitation forecasts generated at the beginning of one season are investigated. For example, the December-January-February (DJF) forecasts are generated at the beginning of December. Similarly, seasonal forecasts for March-April-May (MAM), June-July-August (JJA) and September-October-November (SON)) are respectively produced at the starts of March, June and September.

The concurrent correlation between Niño3.4 and CPC-URD observations is employed to represent ENSO teleconnection (Cai et al., 2009; Kim & Kug, 2018; Steptoe et al., 2018):

$$r(o, \text{Niño3.4}) = \frac{\sum_k (o_k - \overline{o})(\text{Niño3.4}_k - \overline{\text{Niño3.4}})}{\sqrt{\sum_k (o_k - \overline{o})^2} \sqrt{\sum_k (\text{Niño3.4}_k - \overline{\text{Niño3.4}})^2}}.$$

(4)

Niño3.4 is a commonly used index of ENSO (Mason & Goddard, 2001; Chen & Kumar, 2020; Vashisht et al., 2021). The sign of $r(o, \text{Niño3.4})$ indicates the effects of ENSO. A positive $r(o, \text{Niño3.4})$ means that high (low) values of Niño3.4 correspond to large (small) values of observed precipitation, i.e., El Niño events associate with wet conditions whereas La Niña events relate to dry conditions. By contrast, a negative $r(o, \text{Niño3.4})$ indicates that high (low) values of Niño3.4 coincide with below-normal (above-normal) precipitation.

## 3 Methods

### 3.1 Mathematical formulation

The approach to attributing correlation skill of GCM seasonal forecasts to ENSO teleconnection is built upon the coefficient of determination, i.e., $R^2$ (Koster et al., 2010). Mathematically, $R^2$ is equivalent to the squared value of the Pearson's correlation coefficient $r$ (Krause et al., 2005)

$$R^2(Y \sim X) = r^2(Y, X).$$

(5)


There is a difference in the meaning of $r$ in relating observed precipitation to forecasts and ENSO. As to forecasts, $r$ tends to be positive, i.e., high (low) values of forecasts can be indicative of high (low) values of observations (Yuan et al., 2011; Zhao et al., 2020a; Zhao et al., 2020b). However, ENSO teleconnection can be either positive or negative. For example, in DJF, positive $r(o,$ Niño3.4) tends to be dominant over southern North America and negative $r(o,$ Niño3.4) are generally prevalent over northern South America (Mason & Goddard, 2001). Both positive and negative correlations contribute to $R^2$.

Therefore, this paper proposes to use $R^2$ to associate observed precipitation with forecasts and ENSO.

To obtain $R^2$, simple linear regression models are set up to regress observed precipitation against GCM forecasts and Niño3.4 respectively:

$$o_k = \beta_1 f_k + \epsilon_{1,k} \Rightarrow R^2(o \sim f) = 1 - \frac{\sum_k \epsilon_{1,k}^2}{\sum_k (o - \bar{o})^2}, \tag{6}$$

$$o_k = \beta_2 \text{Niño3.4}_k + \epsilon_{2,k} \Rightarrow R^2(o \sim \text{Niño3.4}) = 1 - \frac{\sum_k \epsilon_{2,k}^2}{\sum_k (o - \bar{o})^2}, \tag{7}$$

where $\beta_1$ and $\beta_2$ are the regression coefficients; $\epsilon_1$ and $\epsilon_2$ are the residuals; $k$ represents the target year. Further, through bivariate linear regression, the variance explained by the union of forecasts and Niño3.4 is calculated:

$$o_k = \beta_{3,1} f_k + \beta_{3,2} \text{Niño3.4}_k + \epsilon_{3,k} \Rightarrow R^2(o \sim f \bigcup \text{Niño3.4}) = 1 - \frac{\sum_k \epsilon_{3,k}^2}{\sum_k (o - \bar{o})^2}, \tag{8}$$

in which the union operator is introduced to represent the joint effect. If GCM forecasts were independent from Niño3.4, then $R^2(o\sim f \cup$ Niño3.4) could conceptually be obtained by simply adding up $R^2(o\sim f)$ and $R^2(o\sim$ Niño3.4). On the other hand, if GCM forecasts were dependent on Niño3.4, then there would be some overlaps for $R^2(o\sim f)$ and $R^2(o\sim$ Niño3.4). As a result, $R^2(o\sim f \cup$ Niño3.4) would not be as large as the sum of $R^2(o\sim f)$ and $R^2(o\sim$ Niño3.4).

In accordance with the set theory, the intersection between $R^2(o\sim f)$ and $R^2(o\sim$ Niño3.4) is derived by subtracting
$R^2(o\sim f\cup$ Niño3.4) from the sum of $R^2(o\sim f)$ and $R^2(o\sim$ Niño3.4):

$$\begin{aligned} R^2(o \sim f \bigcap \text{Niño3.4}) = {} & R^2(o \sim f) + R^2(o \sim \text{Niño3.4}) \\ & - R^2(o \sim f \bigcup \text{Niño3.4}) \end{aligned}, \tag{9}$$

in which the intersection operator is introduced to formulate the overlapping $R^2$ between forecasts and Niño3.4. Specifically, the value of $R^2(o\sim f \cap$ Niño3.4) quantifies the overlapping part of the explained variance of observed precipitation accounted for by GCM forecasts and Niño3.4.





**3.2 Attribution of correlation skill**

There are three steps to attribute correlation skill of GCM forecasts to ENSO teleconnection. As shown in Figure 1, the first

step is the implementations of three linear regression models to derive $R^2(o{\sim}f)$, $R^2(o{\sim}\text{Niño3.4})$ and $R^2(o{\sim}f\cup\text{Niño3.4})$ so as to

derive $R^2(o{\sim}f\cap\text{Niño3.4})$. In the absence of a theoretical distribution function for the overlapping $R^2$, the significance is tested

by bootstrapping (Efron, 1979). Specifically, GCM forecasts and Niño3.4 are randomly sampled with replacement for 1000

times under the null hypothesis that observed precipitation is independent from either GCM forecasts or Niño3.4. In this way,

$R^2(o{\sim}f\cap\text{Niño3.4})$ is tested to examine whether the overlapping $R^2$ between $f$ and Niño3.4 is significant.

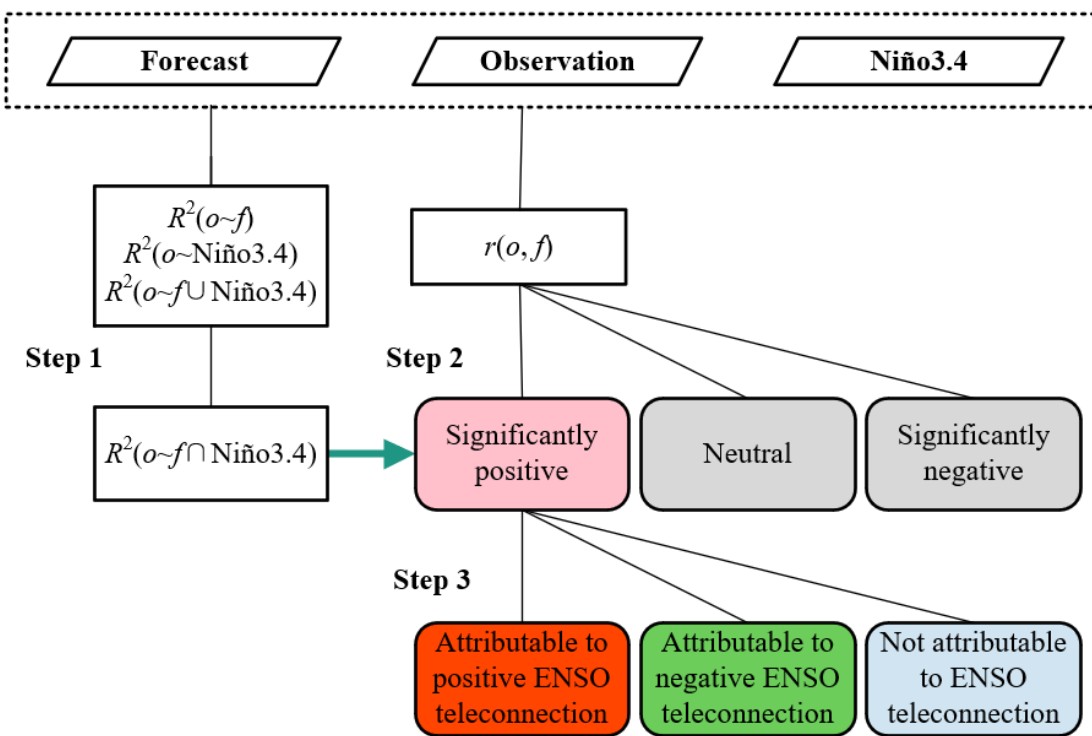

**Figure 1: Flowchart of the three steps that attribute anomaly correlation of GCM forecasts to ENSO teleconnection.**

Secondly, the significance of correlation coefficient is calculated for GCM precipitation forecasts in relating them to global

precipitation. Specifically, $r(o, f)$ is obtained for each grid cell over global land. The two-tailed significance test is

implemented and $r(o, f)$ is therefore identified to be significantly positive, neutral, or significantly negative. In this paper, the

significance levels in the first and second steps are set to be 0.10.

The third step focuses on significantly positive $r(o, f)$ that indicates informative forecasts (Becker et al., 2020; Zhao et al.,

2020a; Zhao et al., 2020b). There are two criteria: (1) the significance of overlapping $R^2$ and (2) the sign of ENSO

teleconnection. Overall, three cases are obtained: (1) significantly positive anomaly correlation attributable to positive ENSO





teleconnection, (2) significantly positive anomaly correlation attributable to negative ENSO teleconnection and (3) significantly positive anomaly correlation not attributable to ENSO teleconnection.

### 3.3 An illustrative example

An example based on synthetic data is devised to illustrate how the overlapping $R^2$ is influenced by the strength of association between two variables. Samples of $x_1$, $x_2$ and $y$ are randomly drawn from a tri-variate normal distribution

$$[x_1, x_2, y]^{\mathrm{T}} \sim N(\boldsymbol{\mu}, \boldsymbol{\Sigma}), \tag{10}$$

where $\boldsymbol{\mu}$ and $\boldsymbol{\Sigma}$ are the mean vector and covariance matrix, respectively:

$$\boldsymbol{\mu}^{\mathrm{T}} = [0,0,0], \tag{11}$$

$$\boldsymbol{\Sigma} = \begin{bmatrix} 1 & r(x_1, x_2) & 0.5 \\ r(x_1, x_2) & 1 & 0.5 \\ 0.5 & 0.5 & 1 \end{bmatrix}. \tag{12}$$

In the example, the correlations of $y$ with $x_1$, $x_2$ are fixed to be 0.5 respectively. As a result, the focus is on $r(x_1, x_2)$ that determines the intersection between $x_1$ and $x_2$. The value of $r(x_1, x_2)$ is set to be 0.0, 0.1, 0.2, 0.3, 0.4 and 0.5. For each pre-155 specified $r(x_1, x_2)$, 1000 samples of $x_1$, $x_2$ and $y$ are drawn to facilitate linear regression models to derive $R^2(y{\sim}x_1)$, $R^2(y{\sim}x_2)$, $R^2(y{\sim}x_1 \cup x_2)$ and $R^2(y{\sim}x_1 \cap x_2)$. For $R^2$, the median and inter-quartile ranges are estimated through 1000 Monte Carlo experiments.



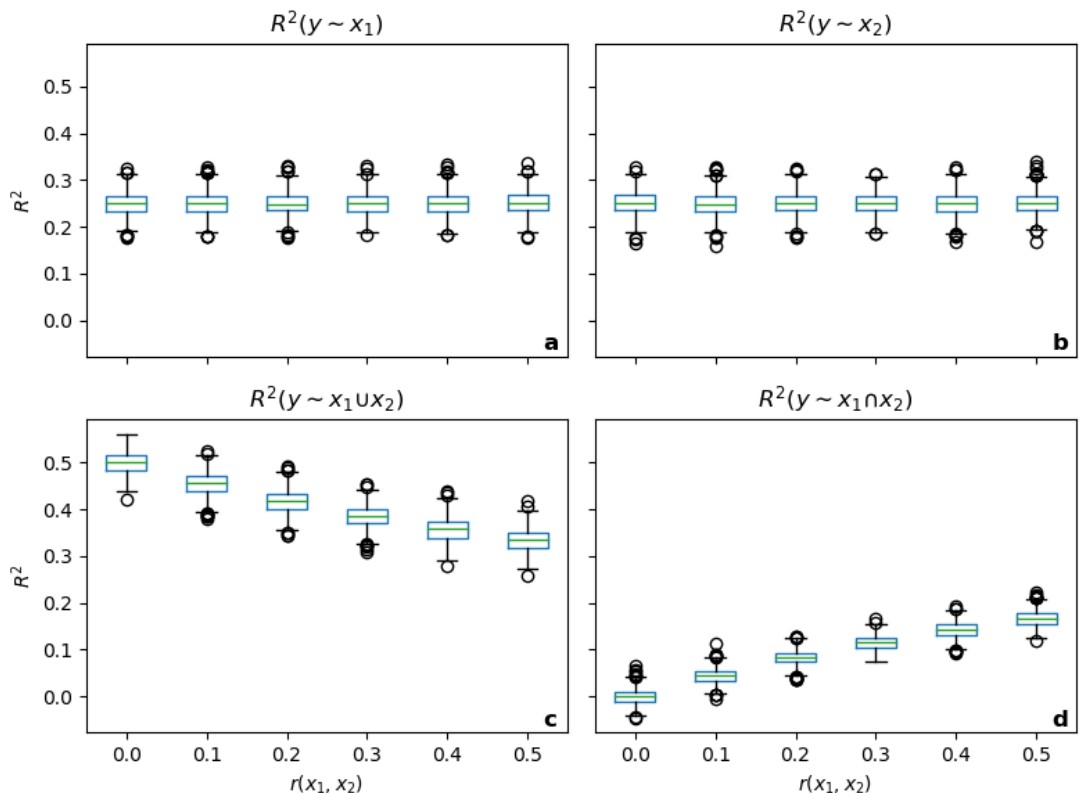

Figure 2: Boxplots of coefficient of determination (R2) for (a) R2(y~x1); (b) R2(y~x2); (c) R2(y~x1 ∪ x2); (d) R2(y~x1∩x2) in the Monte Carlo experiments.





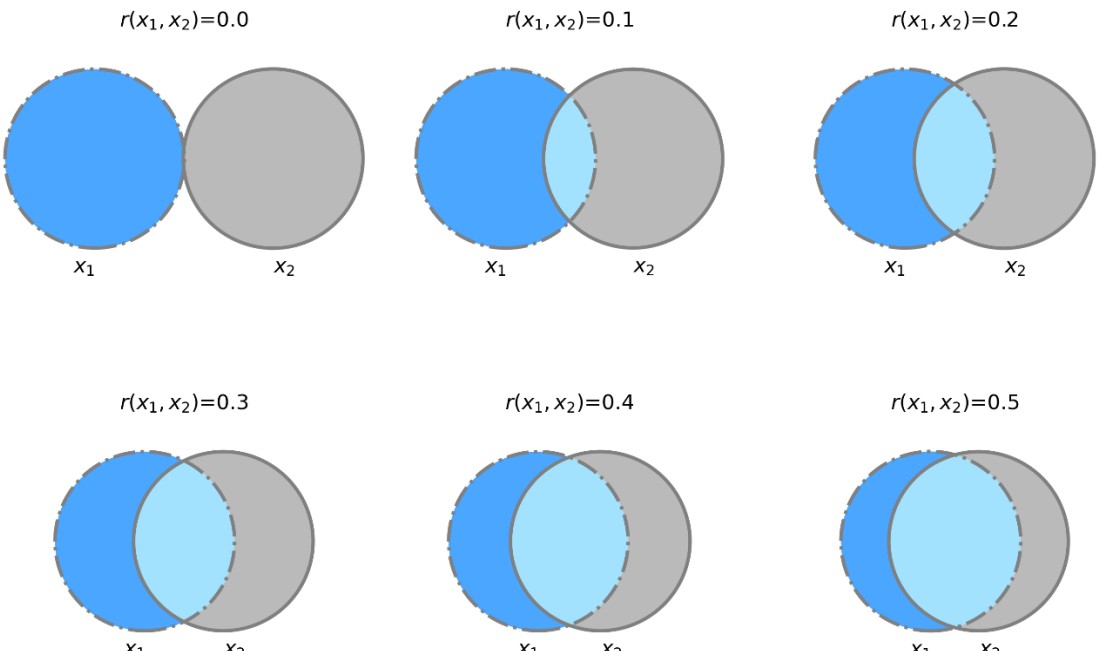

**Figure 3: Venn diagrams for variance of y explained by x1 and x2 given six levels of correlation between x1 and x2.**


Figures 2 and 3 show the influence of $r(x_1, x_2)$ on overlapping $R^2$. According to Figures 2a and 2b, the medians of $R^2(y \sim x_1)$ and $R^2(y \sim x_2)$ approximate the squared value of pre-specified $r(y, x_1)$ and $r(y, x_2)$. They remain to be 0.25, i.e., $0.5^2$, as $r(x_1, x_2)$ increases from 0.0 to 0.5. By contrast, Figure 2c shows that $R^2(y \sim x_1 \cup x_2)$, which represents the ratio of variance explained by the union of $x_1$ and $x_2$, decreases with the increase of $r(x_1, x_2)$. In the meantime, $R^2(y \sim x_1 \cap x_2)$ increases with $r(x_1, x_2)$. Figure 3

further shows the influence of $r(x_1, x_2)$ by using the Venn diagram that illustrates the extent to which $R^2(y \sim x_1)$ and $R^2(y \sim x_2)$ intersect. The area of the circle is proportional to the percentage of variance explained by the corresponding explanatory variable. The intersection is depicted as the overlapping area between the two circles. As is illustrated, it is the correlation between $x_1$ and $x_2$ that leads to the decrease of $R^2(y \sim x_1 \cup x_2)$ and the increase of $R^2(y \sim x_1 \cap x_2)$.

## 4 Results

### 4.1 Correlation skill and ENSO teleconnection in DJF

Global maps of correlation skill and ENSO teleconnection in DJF, which is the peak season of ENSO, are shown in the upper part of Figure 4. In Figure 4a, correlation skill is observed to be largely positive, indicating that CFSv2 forecasts are skillful in general (Saha et al., 2010). In Figure 4b, ENSO teleconnection exhibits both positive and negative values. That is, observed precipitation around the world can be positively or negatively correlated with Niño3.4 (Mason & Goddard, 2001;

Chen & Kumar, 2020; Vashisht et al., 2021). The two-tailed significance test is applied to anomaly correlation and ENSO





teleconnection at each grid cell. Figure 4c illustrates that correlation skill of CFSv2 forecasts are significantly positive over extensive areas around the globe. Also, Figure 4c is observed to correspond to Figure 4d to some extent – significantly positive correlations appear over southern North America and East Africa in both Figures 4c and 4d. In addition, significantly positive anomaly correlation (Figure 4c) correspond to significantly negative ENSO teleconnections (Figure 4d)

in northern South America, southern Africa and Southeast Asia.



**Figure 4: Correlation coefficients between observed precipitation in DJF with (a) CFSv2 forecasts and (b) Niño3.4. Significance tests of correlation for (c) CFSv2 forecasts and (d) Niño3.4. Coefficient of determination (R2) for the regression of observed precipitation against (e) CFSv2 forecasts; (f) Niño3.4; (g) the union of CFSv2 forecasts and Niño3.4 and (h) the intersection of CFSv2 forecasts and Niño3.4.**



The results of linear regressions that lay the foundation for the attribution analysis are shown in Figures 4e to 4h. Figures 4e and 4f, which are respectively for CFSv2 forecasts and Niño3.4, respectively conform to Figures 4a and 4b. This outcome is due to that $R^2$ is mathematically equal to the squared value of correlation coefficient (Krause et al., 2005). The union in Figure 4g exhibits a higher value of $R^2$ than that in either Figure 4e or Figure 4f. The subtraction of the union from the sum facilitates the intersection. As illustrated in Figure 4h, deep blue grid cells are seen to distribute in southern North America, northern South America, East Africa, Southern Africa and Southeast Asia. Over these regions, both GCM forecasts and Niño3.4 index can explain a considerable part of the variance of observed precipitation (Figures 4e and 4f). More importantly, their explained variances intersect (Figure 4h).

## 4.2 Attribution at the global scale

The significance of the intersection for CFSv2 forecasts and Niño3.4 is tested by bootstrapping and then shown in Figure 5. In Figure 5a, grid cells with significant overlapping $R^2$ are marked in orange. The corresponding anomaly correlation and ENSO teleconnection, which are respectively obtained from Figure 4a and Figure 4b, are illustrated using scatter plot in Figure 5b. The scatter points tend fall towards the upper right and left corners of the plot. The implication is that both anomaly correlation and ENSO teleconnection ought to be large enough to facilitate a significant intersection. Largely owing to overlapping $R^2$, anomaly correlation is observed to increase with the increase of positive ENSO teleconnection and also with the decrease of negative ENSO teleconnection. For Figure 5b, there notably exist some outliers that suggest ENSO teleconnection could contribute to negative anomaly correlation. CFSv2 forecasts are generally wrong in these cases and the cautious grid cells are marked in black in Figure 5a. Further, the scatter plot in Figure 5c is for grid cells where overlapping $R^2$ is non-significant. For a fair number of grid cells, anomaly correlation can rise above 0.50 but ENSO teleconnection remains nearly 0.00. The implication is that the corresponding anomaly correlation is not relevant to ENSO teleconnection.



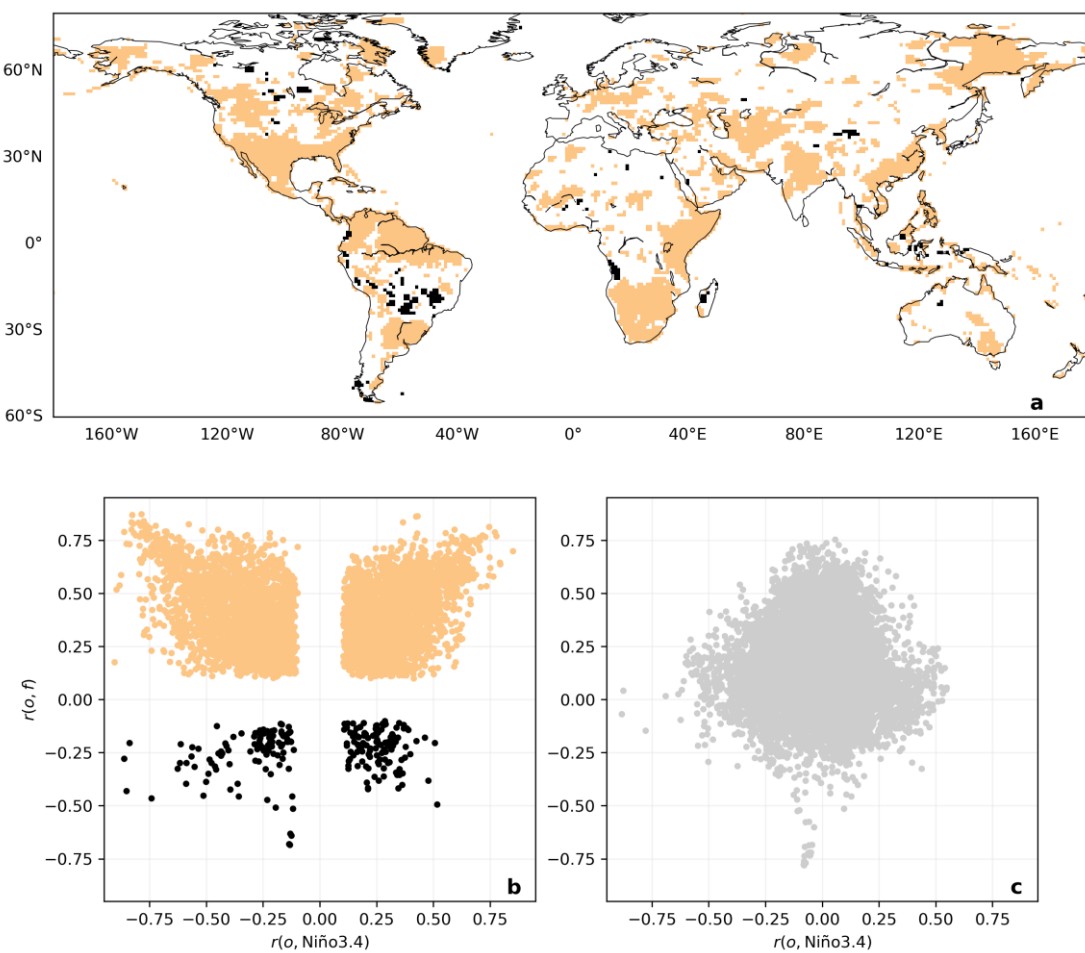

**Figure 5: (a) Spatial distribution of grid cells with significant overlapping R2 for CFSv2 forecasts and Niño3.4 in DJF. Scatter plots of anomaly correlation against ENSO teleconnection for grid cells (b) with significant overlapping R2 and (c) with non-significant overlapping R2.**

Figure 6 presents the three cases of attribution of significantly positive anomaly correlation. The red color marks grid cells

where significantly positive anomaly correlation is attributable to positive ENSO teleconnection. Some corresponding grid cells are observed in regions of known positive ENSO teleconnections, such as southern North America (Strazzo et al., 2019) and equatorial eastern Africa (Vashisht et al., 2021); and some are in less-investigated regions, such as parts of Central, South and East Asia. The green color indicates grid cells where significantly positive anomaly correlation is attributable to negative ENSO teleconnection. They appear in northern South America and southern Africa, where negative ENSO

teleconnection is known to exist (Howard et al., 2019; Cai et al., 2020) and also in parts of Far East and Alaska. There are also gray areas where significantly positive anomaly correlation is not attributable to ENSO teleconnection. The





corresponding grid cells are generally located in Europe, North Asia, northwestern Africa and South Australia. Therein, skillful forecasts can relate to teleconnections other than ENSO, such as AO and NAO (Minami & Takaya, 2020).

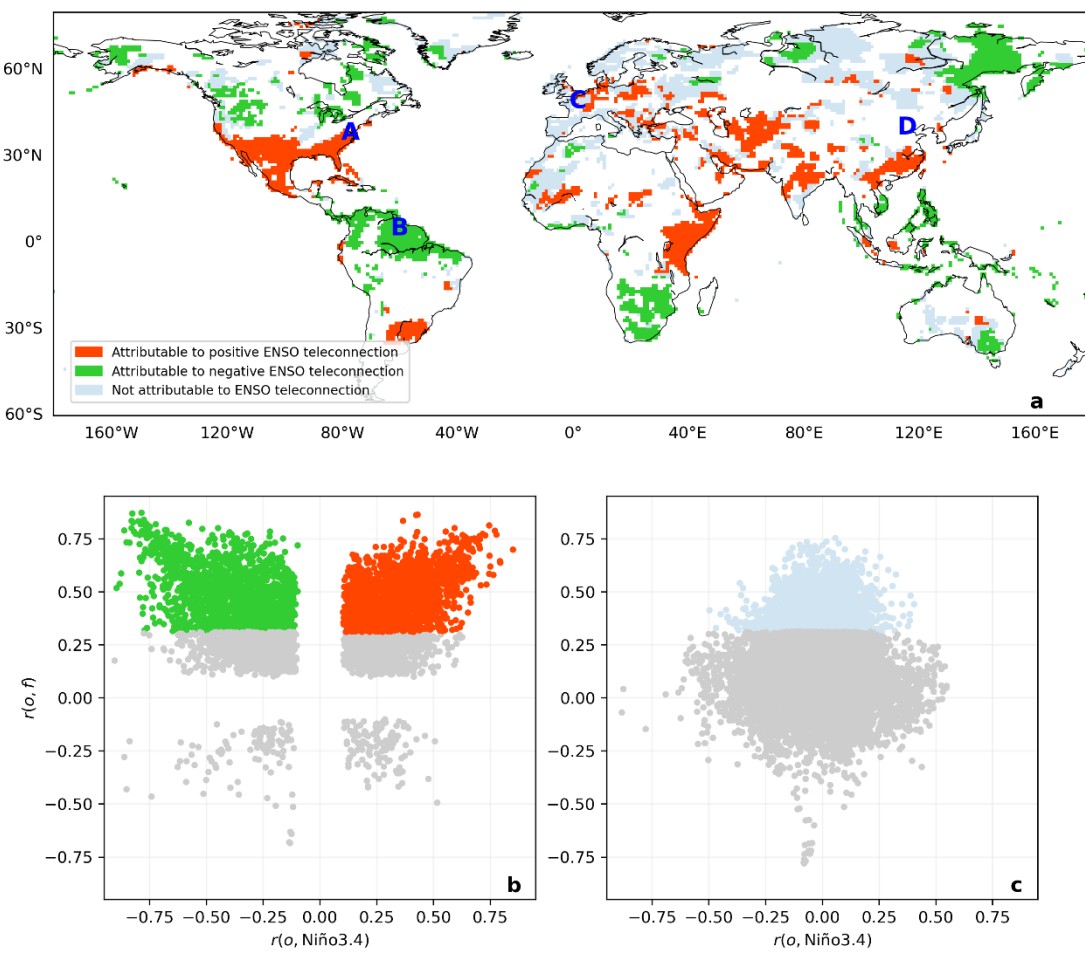

**Figure 6: (a) Spatial distribution of grid cells for the three cases attributing anomaly correlation of CFSv2 forecasts to ENSO teleconnection. Scatter plots of anomaly correlation against ENSO teleconnection for grid cells (b) with significant overlapping R2 and (c) without significant overlapping R2.**

Figure 7 presents a sunburst diagram that quantifies the percentages of significantly positive anomaly correlation and its attribution results. As shown by the central cycle, anomaly correlation is identified to be significantly positive, neutral, or significantly negative (Zhao et al., 2020a; Zhao et al., 2020b). The brown slice suggests that more than half of grid cells around the globe are of neutral anomaly correlation, indicating that GCM precipitation forecasts still have plenty of room for improvement (Kim et al., 2012; Jia et al., 2015; Delworth et al., 2020). The pink slice indicates that 39.4% of the grid cells exhibit significantly positive anomaly correlation. Three cases of attribution are performed for significantly positive anomaly





correlation. The results are shown by the extended slices, of which the color scheme is the same as that of Figure 6. It can be seen that significantly positive anomaly correlation is attributable to positive (negative) teleconnections for 10.8% (11.7%) of grid cells around the globe. Further, significantly positive anomaly correlation is not attributable to ENSO teleconnection for 16.9% of grid cells around the globe.


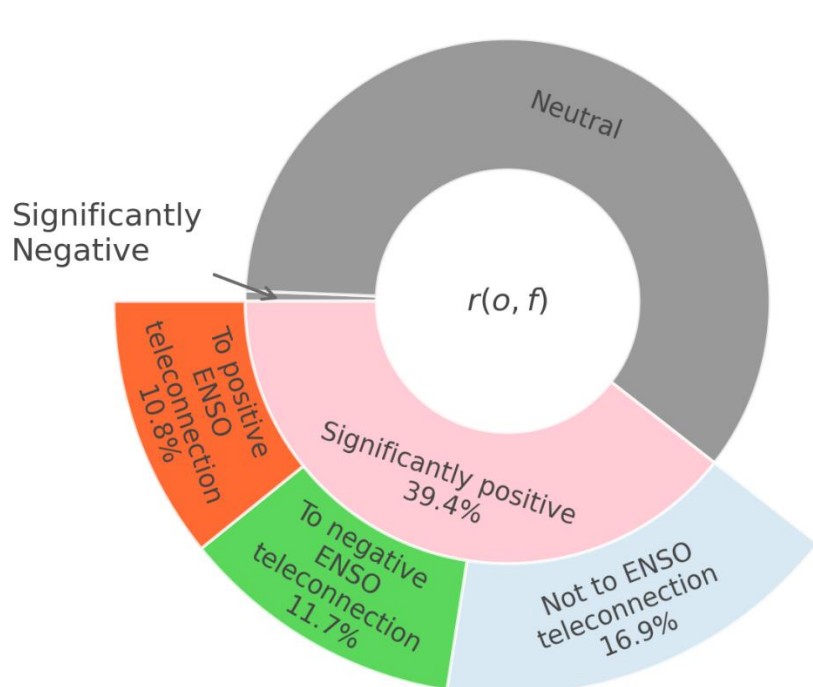

**Figure 7: Sunburst diagram of the attribution of significantly positive anomaly correlation to ENSO teleconnection for CFSv2 forecasts in DJF.**

**4.3 Attribution for selected grid cells**

Four grid cells are selected from Figure 6a to showcase the attribution of CFSv2 forecasts to ENSO teleconnection. As shown in Figure 8, there are three variables under investigation. They are observed precipitation, forecast precipitation and Niño3.4. Their values are normalized, i.e., subtracting the mean and dividing by the standard deviation, to facilitate the inter-comparison of results across the four grid cells. In Figure 8, the first column plots normalized observation against normalized 255 forecasts; the second column plots normalized observation against normalized Niño3.4; and the third column plots





normalized forecasts against normalized Niño3.4. Furthermore, the last column of Figure 8 presents the Venn diagrams illustrating the set operations of union and intersection.

**Figure 8: Scatter plots of the relationships between normalized observation, forecast and Niño3.4 and Venn diagrams of the union and intersection operations for four selected grid cells.**

Grid cell A shown in the first row of Figure 8 presents the case of significantly positive anomaly correlation attributable to positive ENSO teleconnection. The coordinate of the grid cell is (38°N, 77°W). In southern North America, DJF precipitation is known to be modulated by the ENSO-induced Pacific North American (PNA) pattern (Jong et al., 2021). Specifically, in southern North America, PNA tends to cause an enhanced DJF Pacific jet stream that extends further east than normal during





El Niño events and there are nearly reversed patterns during La Niña events . Since the jet stream determines paths of DJF storms, the PNA pattern enables ENSO to affect precipitation in the southern North America. It can be seen that forecasts exhibit a high correlation with Niño3.4. This result suggests that CFSv2 forecasts can reasonably represent the influence of
ENSO. As a result, $R^2$ explained by forecasts and Niño3.4 largely overlap in grid cell A.

Grid cell B shown in the second row of Figure 8 is for the case of significantly positive anomaly correlation attributable to negative ENSO teleconnection. Its coordinate is (5 °N, 60 °W). In DJF, there is a negative ENSO teleconnection over northern South America; it is owing to that ENSO-related SSTs drive changes in the climatological Walker circulation that promotes anomalous descending (ascending) motion and contributes to negative (positive) precipitation anomalies in El Niño (La Niña)
events (Kayano & Andreoli, 2006). The high correlation between forecasts and Niño3.4 highlights the effectiveness of CFSv2 in capturing the negative ENSO teleconnection. There is a considerable intersection between the variability explained by forecasts and Niño3.4 in grid cell B.

Grid cell C displayed in the third row of Figure 8 is for the case of significantly positive correlation not attributable to ENSO teleconnection. Its coordinate is (49 °N, 2 °E). The remote influence of ENSO on Europe has pathways through the North
Atlantic or Arctic regions, including the tropospheric and stratospheric bridges (Butler et al., 2014). However, the amplitude of ENSO impacts is weak and generally not significant in the European region (Butler et al., 2014). Besides, DJF precipitation in Europe is known to be modulated by the NAO (Greuell et al., 2018), the Eurasian snow cover extent and the Quasi-biennial Oscillation (Butler et al., 2014). It can be seen that while observed precipitation shows a neutral correlation with Niño3.4, CFSv2 forecasts explain a substantially fraction of observed precipitation variability. This result indicates the
capability of CFSv2 in capturing teleconnection patterns other than ENSO.

Grid cell D shown in the last row of Figure 8 represents the case of neutral anomaly correlation. The coordinate is (40 °N, 116 °E). Over East Asia, precipitation is known to be influenced by the response of Rossby waves to ENSO (Yang et al., 2018). Also, winter monsoon activities in East Asia are profoundly influenced by wind-SST-evaporation feedbacks over tropical central Pacific to northwestern Pacific (Kim & Kug, 2018). It can be observed that there is a moderate but not
significant correlation between observed precipitation and Niño3.4. However, CFSv2 forecasts exhibit a neutral anomaly correlation, suggesting that the information of ENSO teleconnection is not well represented in the forecasts.

### 4.4 Extended analysis of the other seasons

The attribution analysis is further extended to the other seasons, i.e., MAM, JJA and SON. Global maps of the three cases of attribution are illustrated by season in Figure 9. The results of attribution analysis are observed to vary considerably across
the four seasons (Figures 6, 7 and 9). It is generally owing to the facts that ENSO teleconnection patterns vary by season (Kim & Kug, 2018; Steptoe et al., 2018; Wang et al., 2019) and that GCMs formulate not only ENSO teleconnection but also other teleconnection patterns (Saha et al., 2014; Jia et al., 2015; Delworth et al., 2020). The percentage of significantly anomaly correlation is 27.0%, 24.0% and 34.6% respectively in MAM, JJA and SON. Among them, 7.1%, 6.3% and 7.0% are attributable to positive ENSO teleconnection respectively in MAM, JJA and SON. Representative regions for this case




are western United States in MAM, parts of South America in JJA and Middle East in SON. 7.3%, 7.4% and 14.3% of grid cells are with significantly positive anomaly correlation attributable to negative ENSO teleconnection respectively in MAM, JJA and SON. Representative regions are southeast Asia in MAM, JJA and SON and large parts of Australia in SON. Further, 12.6%, 10.3% and 13.3% of grid cells are with significantly positive anomaly correlation not attributable to ENSO teleconnection. This result calls for the investigation of other teleconnection patterns for GCM seasonal precipitation
forecasts.

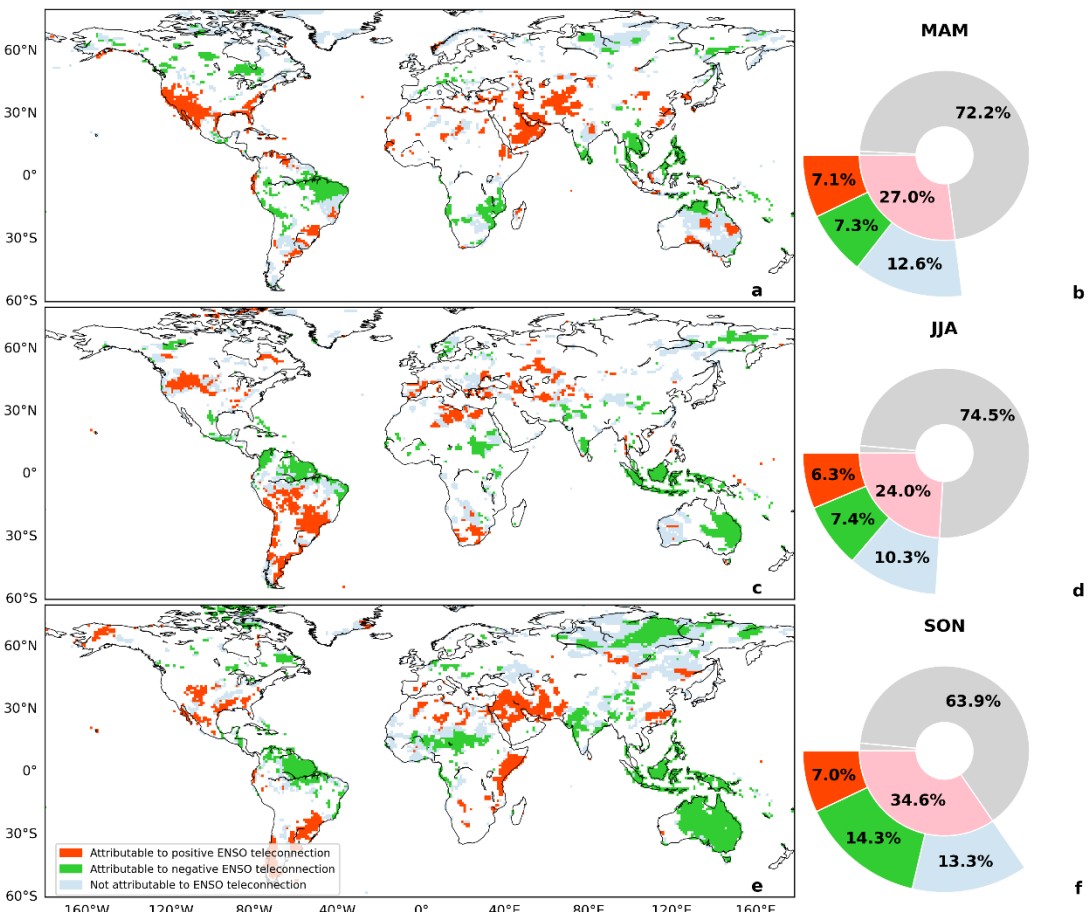

**Figure 9: Spatial maps (left column) and sunburst diagrams (right column) of the three cases of attribution in MAM (upper part), JJA (middle part) and SON (lower part).**






## 5 Discussion

The correlation skill between forecast and observed precipitation is one of the most important indicators of the usefulness of GCM forecasts (Yuan et al., 2011; Becker et al., 2014; Vano et al., 2014; Johnson et al., 2019; Zhao et al., 2020b). To facilitate forecast applications, correlation skill is conventionally calculated from data and then presented using spatial
plotting (Zhao et al., 2020a, 2020b). Focusing on the relationship between correlation skill and ENSO teleconnection, this paper highlights that significantly positive anomaly correlation, which is always advantageous for practical applications of GCM forecasts (Vano et al., 2014; Yuan et al., 2015; Peng et al., 2018), can be attributed to positive (negative) ENSO teleconnection or not to ENSO. In DJF, significantly positive anomaly correlation for CFSv2 forecasts is attributable to positive ENSO teleconnection in southern North America and East Africa and it is attributable to negative ENSO
teleconnection in northern South America and southern Africa. Moreover, significantly positive anomaly correlation in Europe can be attributable to teleconnections other than ENSO. The different cases of attribution also exist for MAM, JJA and SON, but their spatial extents vary considerably. These results conform to previous findings that regions exhibiting positive (negative) ENSO teleconnection change substantially by season (Mason & Goddard, 2001; Chen & Kumar, 2020; Vashisht et al., 2021) and that performances of GCM forecasts vary by season (Vano et al., 2014; Johnson et al., 2019; Zhao
et al., 2020b).

The capability to formulate climate teleconnections is an essential part in the evaluation of GCM forecasts (Molod et al., 2020; Jong et al., 2021). Adding to previous studies that investigated GCM forecasts for regions subject to prominent ENSO influences (Pegion & Kumar, 2013; Manzanas et al., 2014; Jha et al., 2016), this paper presents an investigation of ENSO teleconnection at the global scale. For grid cells around the world, the ratio of significantly positive anomaly correlation
attributable to positive (negative) ENSO teleconnection is respectively 10.8% (11.7%) in DJF, 7.1% (7.3%) in MAM, 6.3% (7.4%) in JJA and 7.0% (14.3%) in SON. Furthermore, the ratio of significantly positive anomaly correlation not attributable to ENSO teleconnection, which suggests that other climate teleconnections are at play in determining the skill of GCM forecasts, is respectively 16.9%, 12.6%, 10.3% and 13.3% in DJF, MAM, JJA and SON. Overall, the spatial plots and the attribution results can serve as a reference for further investigations of the effects of ENSO teleconnections and other climate
patterns on the predictive performance of GCM forecasts.

## 6 Conclusions

Climate teleconnections, in particular ENSO, have been extensively used in conventional statistical hydrological forecasting. This paper is devoted to investigating the relationship between statistical ENSO teleconnection and correlation skill of dynamical CFSv2 forecasts. A novel mathematical approach is built upon the coefficient of determination ($R^2$) that measures
the ratio of explained variance to total variance. Specifically, taking advantage of simple linear regression, the ratios of variance explained by GCM forecasts, Niño3.4 and their union are respectively obtained; then, the overlapping $R^2$ for GCM forecasts and Niño3.4 is derived based on the intersection operation. Based on the significance of overlapping $R^2$ and the



sign of ENSO teleconnection, three cases of attribution are derived. They are significantly positive anomaly correlation attributable to positive ENSO teleconnection, attributable to negative ENSO teleconnection and not attributable to ENSO

teleconnection. The effectiveness of the developed approach is demonstrated through the case study of CFSv2 seasonal forecasts of global precipitation. Spatial plots of the attribution are illustrated by season. The results not only confirm the prominent contributions of ENSO teleconnection to GCM forecasts, but also present spatial plots of regions where significantly positive anomaly correlation is subject to positive ENSO teleconnection, negative ENSO teleconnection and teleconnections other than ENSO. Overall, the attribution approach proposed in this paper can serve as an effective tool to

investigate the source of predictability for GCM seasonal forecasts of global precipitation.

## Acknowledgements

This study is jointly supported by the National Natural Science Foundation of China (51979295, 51861125203 and U1911204) and the Guangdong Provincial Department of Science and Technology (2019ZT08G090).

## Data Availability Statement

All data sets used in this study are publicly available. Both the forecasts and the observations can be downloaded from the International Research Institute for Climate and Society, Earth Institute, Columbia University (https://iridl.ldeo.columbia.edu/SOURCES/.Models/.NMME/).

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
