# Peer review of "Attributing correlation skill of dynamical GCM precipitation forecasts to statistical ENSO teleconnection using a set theory-based approach"

_Hydrology and Earth System Sciences, 2021_

## Author Comment (AC1)

**Response**

*Anonymous Referee #1:*

*Comments: This study evaluated the precipitation forecast skill from dynamical forecast and ENSO teleconnections based on the set theory. The attribution of the forecast skill from the GCM and ENSO for different seasons is separated. Overall this study is well crafted with clear structures.*

Thank you very much for the positive comments.

*Some comments need to be addressed before the potential publication of this study.*

Thank you for the constructive and insightful comments. We have improved the paper accordingly and provide point-by-point responses.

*Lines 61-63: From this statement, it seems that the novelty of this study is the global scale investigation of the ENSO and GCM precipitation forecast. From the following statement (lines 64-65), the attribution of prediction skill seems to appear suddenly. Some improvement in the motivation of this study and associated novelty could enhance the argument.*

Thank you for the constructive comment. The motivation and novelty of this study are illustrated:

"There have been in-depth investigations of ENSO for GCM forecasts as it is one of the most prominent modes of climate variability (Fu et al., 1997; Wang et al., 2003; Feng & Hao, 2021). For example, the influence of ENSO on the East Asian-western Pacific climate was studied for CFSv2 forecasts (Yang & Jiang, 2014) and also for climate projections in the Coupled Model Inter-comparison Project Phase 5 (Gong et al., 2015; Kim et al., 2016; Kim & Kug, 2018). The relationship between forecast skill and the state of sea surface temperatures (SSTs) was evaluated for the seasonal outlook of precipitation over the United States and the skill was found to be dominantly attributed to ENSO in late autumn to late spring (Quan et al., 2006; Pegion & Kumar, 2013; Shin et al., 2019). Understanding the sources for predictability is important for physically-based validations of GCM forecasts (Neelin & Langenbrunner, 2013; Manzanas et al., 2014; Shin et al., 2019). Nevertheless, previous studies tended to pay attention to regions subject to prominent ENSO influences (Kim et al., 2016; Rivera &

Arnould, 2020; Vashisht et al., 2021). At the global scale, the attribution of GCM precipitation forecasts to ENSO teleconnection is yet to be conducted." (Pages 2 to 3, Lines 56 to 66)

"This paper is devoted to attributing correlation skill of dynamical CFSv2 forecasts (e.g., Saha et al., 2010; Yuan et al., 2011; Jia et al., 2015; Becker et al., 2020; Zhao et al., 2020a) to statistical ENSO teleconnection at the global scale. A novel approach based on the set theory is devised to facilitate the attribution. With the coefficient of determination ($R^2$) characterizing the ratio of explained variance and the set operations illustrating the overlapping $R^2$, the significance test by grid cell is conducted by bootstrapping in order to identify where correlation skill is attributable to ENSO teleconnection. As will be demonstrated through the case study of CFSv2 forecasts, three cases are effectively revealed: (1) significantly positive anomaly correlation attributable to positive ENSO teleconnection, (2) significantly positive anomaly correlation attributable to negative ENSO teleconnection and (3) significantly positive anomaly correlation not attributable to ENSO teleconnection." (Page 3, Lines 67 to 74)

*Line 115: "associate P to forecast and ENSO"? This is not clear. Please clarify or revise.*

Thank you for the comment. We have rephrased it in the revision:

"Both positive and negative correlations contribute to $R^2$. $R^2$ is respectively derived for forecasts and ENSO in relating them to observed precipitation." (Page 5, Lines 117 to 119)

*Figure 2. In this figure, the R2(y~x1+x2) decrease with r(x1,x2). What is the meaning in the context of the GCM forecast and ENSO? (the coefficient of determination decreases with the correlation between GCM and ENSO?)*

Thank you. The paragraph mixed Figures 2 and 3, which could be confusing. In the revision, Figures 2 and 3 are illustrated in two paragraphs.

[Figure]

Figure 2: Boxplots of coefficient of determination ($R^2$) for (a) $R^2(y\sim x_1)$, (b) $R^2(y\sim x_2)$, (c) $R^2(y\sim x_1 \cup x_2)$ and (d) $R^2(y\sim x_1 \cap x_2)$ in the Monte Carlo experiments.

"Figures 2 illustrates the influences of $r(x_1, x_2)$ on $R^2(y\sim x_1)$, $R^2(y\sim x_2)$, $R^2(y\sim x_1 \cup x_2)$ and $R^2(y\sim x_1 \cap x_2)$. Figures 2a and 2b show that the median values of $R^2(y\sim x_1)$ and $R^2(y\sim x_2)$ approximate the squared value of pre-specified $r(y, x_1)$ and $r(y, x_2)$. That is, they remain to be around 0.25, i.e., $0.5^2$, as $r(x_1, x_2)$ increases from 0.0 to 0.5. The indication is that the change in correlation between forecasts and Niño3.4 does not influence the amount of information that they respectively provide for observed precipitation. By contrast, Figure 2c shows that $R^2(y\sim x_1 \cup x_2)$, which represents the ratio of variance explained by the union of $x_1$ and $x_2$, decreases with the increase of $r(x_1, x_2)$. This phenomenon coincides with the increase of $R^2(y\sim x_1 \cap x_2)$ with $r(x_1, x_2)$." (Page 8, Lines 167 to 172)

[Figure]

Figure 3: Venn diagrams for the variance of y explained by $x_1$ and $x_2$ given six levels of correlation between $x_1$ and $x_2$.

  "Figure 3 further shows the influence of $r(x_1, x_2)$ by using the Venn diagram that illustrates the extent to which $R^2(y{\sim}x_1)$ and $R^2(y{\sim}x_2)$ intersect. The intersection is represented by the overlapping area between the two circles. From this figure, it can be seen that it is the correlation between $x_1$ and $x_2$ that leads to the decrease of $R^2(y{\sim}x_1 \cup x_2)$ and the increase of $R^2(y{\sim}x_1 \cap x_2)$. For global precipitation forecasting, the intersection reflects the overlapping $R^2$ for GCM forecasts and ENSO teleconnection. Figure 3 indicates that the correlation between GCM forecasts and Niño3.4 would lead to a decrease of total information and an increase of overlapping information." (Page 9, Lines 177 to 182)

*Lines 177-178: What is the lead time of this figure 4a (and the whole study)? In addition, please add the title or notation of the Figure 4(c-d) to make it clear.*

Thank you. The lead time is 0 month for the results presented in this figure:

"The attention is paid to the latest forecasts. That is, seasonal precipitation forecasts generated at the beginning of the season are investigated. For example, the December-January-February (DJF) forecasts are generated at the beginning of December. Similarly, seasonal forecasts for March-April-May (MAM), June-July-August (JJA) and September-October-November (SON)) are respectively produced at the starts of

March, June and September." (Page 4, Lines 96 to 99)

Moreover, we have added subtitles for Figure 4(c-d) and provided more details in the figure caption:

[Figure]

"

Figure 4: Correlation coefficients between observed precipitation in DJF with (a) 0-month lead CFSv2 forecasts generated at the beginning of December and (b) concurrent Niño3.4. Significance tests of correlation for (c) CFSv2 forecasts and (d) Niño3.4. Coefficient of determination ($R^2$) for the regression of observed precipitation against (e) CFSv2 forecasts; (f) Niño3.4; (g) the union of CFSv2 forecasts and Niño3.4 and (h) the intersection of CFSv2 forecasts and Niño3.4." (Page 11, Lines 204 to 208)

*Figure 8: The Venn diagrams for grid cells A and B seem to be close and are hard to distinguish.*

Thank you for the comment. We have double checked the results:

"It is noted that some similarities in the Venn diagrams are observed for grid cells A and B at which ENSO teleconnection is respectively positive and negative. Specifically, $r(o, Niño3.4)$ is respectively 0.46 and –0.48 at grid cells A and B. $R^2$ that is the focus of the Venn diagrams mathematically represents the squared value of correlation coefficient. Therefore, the similar Venn diagrams highlight that both positive and negative ENSO teleconnection can contribute to correlation skill." (Page 17, Lines 286 to 290)

*Section 4.4. These patterns are interesting. Have you compared with other studies and see if these patterns are consistent with previous studies?*

Thank you for the constructive comment. These patterns are consistent with previous findings. More illustrations are added:

"The attribution analysis is further extended to the other seasons, i.e., MAM, JJA and SON. Global maps of the three cases of attribution are illustrated by season in Figure 9 and also in Figures S1 to S6 of the supplementary material. Overall, the results of attribution vary considerably across the four seasons. It is generally owing to the facts that ENSO teleconnection varies by season (Kim & Kug, 2018; Steptoe et al., 2018; Wang et al., 2019) and that GCMs formulate not only ENSO but also other teleconnections (Saha et al., 2014; Jia et al., 2015; Delworth et al., 2020). Overall, the percentage of significantly positive anomaly correlation is 27.0%, 24.0% and 34.6% respectively in MAM, JJA and SON, which tend to be smaller than that in DJF. This result can be due to the seasonal cycle of ENSO, i.e., ENSO forcing tends to the strongest in DJF and it translates into weaker precipitation variability in the other seasons (Yang et al., 2018).

The percentage of significantly positive anomaly correlation attributable to positive ENSO teleconnection is respectively 7.1%, 6.3% and 7.0% in MAM, JJA and SON. Representative regions for this case are western United States in MAM (Pegion & Kumar, 2013), parts of South America in JJA (Cai et al., 2020) and Middle East in SON (Mariotti, 2007). 7.3%, 7.4% and 14.3% of grid cells are with significantly positive anomaly correlation attributable to negative ENSO teleconnection respectively in MAM, JJA and SON. One representative region is southeast Asia, where precipitation is strongly correlated with ENSO in MAM, JJA and SON (Jiang & Li, 2017); also, in Australia, precipitation in SON is found to be substantially influenced by the extratropical teleconnection pathway of ENSO (Cai & Weller, 2013). Furthermore, 12.6%, 10.3% and 13.3% of grid cells are with significantly positive anomaly

correlation not attributable to ENSO teleconnection. This result calls for the investigation of other teleconnection patterns for GCM seasonal precipitation forecasts." (Pages 17 to 18, Lines 304 to 322)

---

## Author Comment (AC2)

**Response**

*Anonymous Referee #2*:

*This work develops a statistical approach to attributing correlation skill of dynamical forecast to ENSO teleconnection. It can present regions where the forecast skill is attributed to its teleconnections with ENSO, can serve as an effective tool to investigate the source of predictability. The method and results sound reasonable.*

Thank you very much for the positive comments.

*It is potentially publication if the following concerns are included, regarding to the lead-lag teleconnections and forecast.*

We are grateful to you for the insightful and constructive comments. Please see the point-by-point responses in the following.

1. *This approach is just applied with concurrent correlation between Nino3.4 index and observations, to represent ENSO teleconnection. But ENSO also has some lead-lag impacts on the precipitation variations, which also involve in the forecast skill and sources of precipitation. How about of the attributions of these processes? More discussions on it are needed.*

Thank you for the comment. New experiments are performed to investigate lagged ENSO teleconnection:

"Lagged ENSO teleconnection and correlation skill by lead time are important issues for seasonal forecasting (Schepen et al., 2012; Peng et al., 2014; Steinschneider & Lall, 2016). A numerical experiment of attribution is performed for Niño3.4 indices at different time lags. Specifically, for precipitation in DJF, the concurrent Niño3.4 in the analysis is replaced by Niño3.4 in November, October and September so as to investigate the overlapping $R^2$ for 1-, 2- and 3-month lag ENSO teleconnection. Figures S7 to S10 in the supplementary material show that the results tend to be similar at the three lags; the similarities are generally owing to the temporal persistency of Niño3.4 (Yang et al., 2018). Furthermore, another experiment is devised for GCM forecasts at the lead time of 1, 2 and 3 months. That is, for precipitation in DJF, forecasts generated in November, October and September are used to replace forecasts generated in

December in the analysis (Figures S11 to S14). It can be observed that the case of significantly positive anomaly correlation attributable to positive ENSO teleconnection remains for southern North America and that the case of significantly positive anomaly correlation attributable to negative ENSO teleconnection remains for northern South America and southern Africa." (Page 20, Lines 343 to 353)

"

Figure S10: As in Figure 9 but for DJF seasonal precipitation with (a-b) November (1-month lag), (c-d) October (2-month lag) and (e-f) September (3-month lag) Niño3.4 index."

2. *Lines 20-24, and 346-350, for the novelty of this work in Abstract and Conclusion, there is a repetition in writing. Please rewrite them.*

Thank you. We have rewritten the Conclusion:

"The spatial patterns of forecast skill attributed to different types of ENSO teleconnections confirm previous studies associating seasonal precipitation variability with ENSO and highlight the capability of CFSv2 in capturing the pathways of ENSO

teleconnections. The attribution method proposed in this paper can lay a basis for future evaluations of other teleconnections and investigations of predictability sources for GCM seasonal precipitation forecasts." (Page 20, Lines 374 to 378)

3. *Line 94, this work is just "paid to the latest forecasts". It is also quite interesting to figure out that, are there any differences on the ratio of significantly positive anomaly correlation attributable to different types of ENSO teleconnections, with the increase of lead time?*

Thank you for the constructive comment. New experiments are performed to investigate the effects of lead time:

"Lagged ENSO teleconnection and correlation skill by lead time are important issues for seasonal forecasting (Schepen et al., 2012; Peng et al., 2014; Steinschneider & Lall, 2016). A numerical experiment of attribution is performed for Niño3.4 indices at different time lags. Specifically, for precipitation in DJF, the concurrent Niño3.4 in the analysis is replaced by Niño3.4 in November, October and September so as to investigate the overlapping R2 for 1-, 2- and 3-month lag ENSO teleconnection. Figures S7 to S10 in the supplementary material show that the results tend to be similar at the three lags; the similarities are generally owing to the temporal persistency of Niño3.4 (Yang et al., 2018). Furthermore, another experiment is devised for GCM forecasts at the lead time of 1, 2 and 3 months. That is, for precipitation in DJF, forecasts generated in November, October and September are used to replace forecasts generated in December in the analysis (Figures S11 to S14). It can be observed that the case of significantly positive anomaly correlation attributable to positive ENSO teleconnection remains for southern North America and that the case of significantly positive anomaly correlation attributable to negative ENSO teleconnection remains for northern South America and southern Africa." (Page 20, Lines 343 to 353)

"

[Figure]

Figure S14: As in Figure 9 but for DJF seasonal forecasts generated in (a-b) November (1-month lead), (c-d) October (2-month lead) and (e-f) September (3-month lead)."

4. *For the extended analysis in Section 4.4, the percentage of significantly anomaly correlation in MAM, JJA and SON is largest in SON, but still lower than DJF. It suggests obvious seasonal differences. This may connect with the observed teleconnections with ENSO. It would be better giving more discussions on these differences and more comparisons with previous sections and studies.*

Thank you for the insightful comment. More discussions and comparisons are added in Section 4.4:

"The attribution analysis is further extended to the other seasons, i.e., MAM, JJA and SON. Global maps of the three cases of attribution are illustrated by season in Figure 9 and also in Figures S1 to S6 of the supplementary material. Overall, the results of attribution vary considerably across the four seasons. It is generally owing to the facts that ENSO teleconnection varies by season (Kim & Kug, 2018; Steptoe et al., 2018; Wang et al., 2019) and that GCMs formulate not only ENSO but also other

teleconnections (Saha et al., 2014; Jia et al., 2015; Delworth et al., 2020). Overall, the percentage of significantly positive anomaly correlation is 27.0%, 24.0% and 34.6% respectively in MAM, JJA and SON, which tend to be smaller than that in DJF. This result can be due to the seasonal cycle of ENSO, i.e., ENSO forcing tends to the strongest in DJF and it translates into weaker precipitation variability in the other seasons (Yang et al., 2018).

The percentage of significantly positive anomaly correlation attributable to positive ENSO teleconnection is respectively 7.1%, 6.3% and 7.0% in MAM, JJA and SON. Representative regions for this case are western United States in MAM (Pegion & Kumar, 2013), parts of South America in JJA (Cai et al., 2020) and Middle East in SON (Mariotti, 2007). 7.3%, 7.4% and 14.3% of grid cells are with significantly positive anomaly correlation attributable to negative ENSO teleconnection respectively in MAM, JJA and SON. One representative region is southeast Asia, where precipitation is strongly correlated with ENSO in MAM, JJA and SON (Jiang & Li, 2017); also, in Australia, precipitation in SON is found to be substantially influenced by the extratropical teleconnection pathway of ENSO (Cai & Weller, 2013). Furthermore, 12.6%, 10.3% and 13.3% of grid cells are with significantly positive anomaly correlation not attributable to ENSO teleconnection. This result calls for the investigation of other teleconnection patterns for GCM seasonal precipitation forecasts." (Pages 17 to 18, Lines 304 to 322)

---

## Author Comment (AC3)

**Response**

*Anonymous Referee #3:*

*Referee comment on "Attributing correlation skill of dynamical global precipitation forecasts to statistical ENSO teleconnection using a set theory based approach" by Tongtiegang Zhao et al., Hydrol. Earth Syst. Sci. Discuss., https://doi.org/10.5194/hess-2021-328-RC3, 2021*

*None.*

Thank you. We have conducted a thorough revision to improve the paper.